# Overexpression of the β-Subunit of Acid Ceramidase in the Epidermis of Mice Provokes Atopic Dermatitis-like Skin Symptoms

**DOI:** 10.3390/ijms25168737

**Published:** 2024-08-10

**Authors:** Miho Sashikawa-Kimura, Mariko Takada, Md Razib Hossain, Hidetoshi Tsuda, Xiaonan Xie, Mayumi Komine, Mamitaro Ohtsuki, Genji Imokawa

**Affiliations:** 1Department of Dermatology, Jichi Medical University, 3311-1 Yakushiji, Shimotsuke 329-0498, Tochigi, Japan; sashikawa@jichi.ac.jp (M.S.-K.); razib@jichi.ac.jp (M.R.H.); thidet@jichi.ac.jp (H.T.); mamitaro@jichi.ac.jp (M.O.); 2Center for Bioscience Research and Education, Utsunomiya University, 350 Mine, Utsunomiya 321-8505, Tochigi, Japan; mrktkd0711@cc.utsunomiya-u.ac.jp (M.T.); xie@cc.utsunomiya-u.ac.jp (X.X.)

**Keywords:** atopic dermatitis, acid ceramidase beta-subunit, transgenic mice, sphingomyelin/glucosylceramide deacylase, IL-33, macrophage accumulation

## Abstract

We previously reported that a pathogenic abnormality in the barrier and water-holding functions of the stratum corneum (SC) in the skin of patients with atopic dermatitis (AD) is mainly attributable to significantly decreased levels of total ceramides in the SC. That decrease is mediated by the abnormal expression of a novel ceramide-reducing enzyme, sphingomyelin/glucosylceramide deacylase (SGDase), which is the β-subunit (ASAH1b) of acid ceramidase. In this study, we determined whether mice overexpressing ASAH1b in their epidermis develop AD-like skin symptoms. We generated transgenic (TG) mice overexpressing ASAH1b, regulated by the involucrin promoter, to localize its expression in the upper epidermis. After hair removal using a depilatory cream containing glycolic acid, the TG mice without any visible skin inflammation at 8 weeks of age had increased levels of ASAH1b and decreased levels of SC ceramide, with disrupted barrier functions measured by trans-epidermal water loss compared to the wild-type (WT) mice. Interestingly, enzymatic assays revealed that SGDase activity was not detectable in the skin of the TG mice compared to WT mice. Immunological staining revealed that there was an increased expression level of IL-33 in the epidermis and an accumulation of macrophages in the dermis of TG mice compared to WT mice, which are phenotypic characteristics of AD, that were exacerbated by tape-stripping of the skin. In the skin of the TG mice, the mRNA levels of IL-5, CCL11, IL-22, CXCL10, and IFNγ were significantly upregulated compared to the WT mice, and tape-stripping significantly increased the mRNA levels of IL-4, IL-33, CXCL1, CXCL12, TLR9, and CD163 compared to WT mice. These findings strongly indicate that the skin of the depilatory cream-treated TG mice exists in an atopic dry skin condition that is highly sensitive to various environmental stimuli. The sum of our results suggests that ASAH1b itself, even in the absence of its enzymatic activity, is a major etiologic factor for atopic dry skin symptoms via an unknown mechanism.

## 1. Introduction

Atopic dermatitis (AD) is characterized clinically by severe dryness, itchiness, and immunological abnormalities of the skin as well as physiologically by increased cutaneous permeability and deficient water-holding functions of the stratum corneum (SC) even in non-lesional skin, which is called atopic dry skin [1,2,3,4,5,6,7,8,9,10,11,12]. The barrier dysfunction of the SC in the skin of AD patients is attributable to a high vulnerability to irritants and/or allergens, the sensitization of which under the disrupted barrier conditions causes a Th2 type immunological abnormality characterized mainly by increased levels of IgE and the infiltration of eosinophils [13,14,15]. The deficient water-holding functions in AD produces severe dry skin [16] that is a prerequisite factor for the easily provoked itching. Thus, the clinical and physiological features of atopic dry skin serve as etiological factors that trigger recurrent dermatitis [7]. Many studies have demonstrated that the SC dysfunction of AD patients is due to the significantly decreased levels of total ceramides in the SC [7,17]. This pathophysiological relationship is strengthened by evidence that ceramides can act as water modulators by incorporating bound water molecules [7] and also as a permeability barrier [7] into and through multi-layered lamellar structures between cells in the SC.

The essential role of the ceramide deficiency in the pathogenesis of AD has been corroborated by clinical studies [7], demonstrating that the repeated topical application of a synthetic pseudo-ceramide (pCer) to AD skin significantly ameliorates the atopic dry skin symptoms, accompanied by recovery of the SC barrier/water reservoir functions. The detailed analysis of ceramide profiles in our clinical study [18] revealed that these improvements in atopic dry skin and the functional amelioration of the SC can be attained without any recovery of the downregulated levels of total endogenous ceramides but with the applied pCer compensating for the deficiency of endogenous ceramides in the SC. The remaining levels of applied pCer, but not any ceramide species in the SC of AD skin, are paralleled by the increased water content in AD skin. These findings strongly support the hypothesis that total ceramide levels, including the pCer that penetrates and accumulates in the SC, play a more critical role in modulating the barrier and water reservoir functions than the differential ceramide profiles, resulting in the improvement of atopic dry skin symptoms. It seems likely that the amelioration of atopic dry skin is an essential requirement to prevent recurrent dermatitis in AD patients.

To characterize the biological mechanism(s) involved in the pathogenesis of AD, it was important to elucidate what biological factor(s) cause the downregulation of ceramides in the SC of AD skin. Three sphingolipid hydrolysis enzymes, β-glucocerebrosidase (GBA), acid sphingomyelinase (aSMase) and acid ceramidase (ASAH1), function at the interface between the stratum granulosum and the SC, following the secretion of lamellar granules (LGs), which modulate ceramide levels in the SC. ASAH1 and aSMase, the downregulation of which directly leads to a decrease in SC ceramides, are not downregulated at the enzymatic activity level [7] or at the protein level [19,20] in non-lesional epidermis from AD skin, although one study reported a decreased activity of aSMase [21]. Further, other sphingolipid metabolic enzymes that are associated with the synthesis of glucosylceramide (GCer) and sphingomyelin (SM), such as serine-palmitoyl transferase (SPT), stearoyl CoA desaturase (SCD) [20], ceramide synthases (CERS) 1–5 [20], GCer synthase (GCERS), alkyl chain elongation enzymes [20] and SM synthase (SMS), have never been implicated in the abnormal ceramide deficiency of non-lesional AD skin.

To this end, as an etiologic and essential bio-factor that can provoke ceramide deficiency even during the normal ceramide metabolism observed in AD skin, we discovered a novel enzyme, sphingomyelin/glucosylceramide deacylase (SGDase), which cleaves the N-acyl linkage of SM and GCer [7]. SGDase acts by competing with aSMase and ASAH1 to yield their lysoforms, sphingosylphosphorylcholine (SPC) and glucosylsphingosine (GS), respectively, instead of ceramide, which results in the ceramide deficiency in AD skin. SGDase activity is significantly upregulated in AD skin including the lesional epidermis as well as the involved and uninvolved SC, but not in the skin of patients with contact dermatitis or chronic eczema, compared with healthy controls (HCs) [7]. Those reaction products (SPC and GS) occur to a greater extent in the involved and uninvolved SC of AD skin compared with chronic eczema or contact dermatitis, as well as HCs with significant correlations of the increased levels of SPC and GS with the decreased levels of total ceramides or acylceramide (Cer[EOS]) [7]. Further, we hypothesized a possible role of SPC in the clinical features of atopic dry skin including the pigmented skin, roughened and hardened SC cells, and epidermis that are susceptible to allergens. To test that hypothesis, we demonstrated the stimulatory effects of SPC on pigmentation by melanocytes [7] and on ICAM-1 expression [7] and keratinization (transglutaminase 1 expression) [7] by keratinocytes. Taken together, those studies suggest that the two typical physiological features of atopic dry skin, the water-holding and barrier dysfunctions, are attributable to the ceramide deficiency in the SC that could be induced by the enzymatic action of abnormally provoked SGDase.

Therefore, it was important to purify SGDase to homogeneity and characterize it at the gene and protein levels. Fortunately, SGDase from rat skin had recently been purified to homogeneity and had an apparent molecular mass of 43 kDa, and maximal pH and pI values of 5.0 and around 7.0, respectively [22]. The purified rat SGDase followed normal Michaelis–Menten kinetics with a V_max_ of 14.1 nmol/mg/h and a K_m_ of 110.5 µM. Those properties of pH dependency, size, and kinetic parameters were identical to the characteristics of the enzyme observed in our study of AD skin [23]. Importantly, we were able to conclude that SGDase is identical to the β-subunit of ASAH1, which consists of α- and β-subunits linked by a single S–S bond, the disruption of which results in activating the catalytic function of SGDase [22]. That discovery prompted us to create a novel animal model for AD, transgenic (TG) mice that have an upregulated expression of the β-subunit of ASAH1 controlled by the involucrin promoter (ASAH1b TG mice), which should express ASAH1b exclusively in the upper layers of the epidermis. The results of our study clearly show that the overexpression of ASAH1b in the upper epidermis causes the spontaneous development of AD-resembling skin reactions in TG mice, even though SGDase activity was not detectable in the epidermis of those mice. These findings strongly indicate that the skin of 8-week-old TG mice even without any visible skin inflammation predominantly exists in an atopic dry skin condition that is highly sensitive to various environmental stimuli. The sum of our results suggests that ASAH1b itself, regardless of its enzymatic activity, is a major etiologic factor for atopic dry skin symptoms although the precise mechanism involved remains obscure.

## 2. Results

### 2.1. Overexpression of ASAH1b in the Upper Epidermis of the Ear Skin of TG Mice Facilities the Susceptibility to Tape-Stripping

The ear skin of the wild-type (WT) and of TG mice at the age of 8 weeks without any hair removal treatment did not have any visible skin symptoms, including scaling (Figure 1A). However, repeated tape-stripping of both the inside and the outside of the ear skin induced scaly dermatitis, and the TG mice had more severe dermatitis than the WT mice (Figure 1A,B), accompanied by a significantly increased ear thickness compared to WT mice (Figure 1B). These findings suggest a higher vulnerability against stimuli in the ear skin of TG mice compared to WT mice.

### 2.2. Overexpression of ASAH1b in the Upper Epidermis Does Not Elicit Barrier-Disruption under Normal Conditions

Three days after hair removal using electric hair clippers, the dorsal skin of the TG and WT mice at the age of 8 weeks did not have any visible skin symptoms including scaling (Figure 2A), accompanied by similar TEWL values (Figure 2B), which strongly suggests that there is no barrier disruption in the skin of the TG mice under normal conditions compared to the WT mice.

### 2.3. Overexpression of ASAH1b in the Upper Epidermis Causes the Spontaneous Development of Skin Reactions in TG Mice That Resemble AD after Hair Removal Using a Depilatory Cream Containing Glycolic Acid

Three days after hair removal using a depilatory cream containing glycolic acid, the dorsal skin of 8-week-old TG mice without any visible skin inflammation on the non-tape-stripped left side exhibited a distinct dryness with fine scaling, whereas the dorsal skin of the WT mice did not have any visible skin symptoms including scaling (Figure 3A). The dorsal skin of the TG mice exhibited a significant increase in TEWL value compared to WT mice (Figure 3B), which strongly suggests that the barrier disruption occurs without the development of skin inflammation in TG mice. H&E staining of the dorsal skin of TG mice revealed a thickened epidermis with mild hyperkeratosis (Figure 4A), accompanied by a significantly increased epidermal thickness (Figure 4B), compared to WT mice.

### 2.4. Overexpression of ASAH1b in the Upper Epidermis Facilities the Susceptibility against Tape-Stripping in the Dorsal Skin of Mice

To characterize differences in skin reactions between the WT and TG mice, 1 day after hair removal using a depilatory cream, the right side of the dorsal skin of each mouse was tape-stripped, with the left side not being tape-stripped. Two days later, the dorsal skin of TG mice exhibited scaly dermatitis more severely than WT mice (Figure 3A). Skin specimens from the tape-stripped areas of TG mice and WT mice were stained with H&E, which demonstrated that although tape-stripping did not increase the thickness of the epidermis, TG mice still had a thicker epidermis compared to WT mice even after tape-stripping (Figure 4). The tape-stripping significantly upregulated TEWL values in the WT and in TG mice, with the TEWL values remaining slightly increased in TG mice compared to WT mice (Figure 3B).

### 2.5. Immunostaining with an Anti-ASAH1b Antibody and an Anti-Ceramide Antibody before and after Tape-Stripping and Western Blotting of ASAH1b Protein in the Upper Epidermis of TG Mice Overexpressing ASAH1b

Immunostaining with an anti-ASAH1b antibody and an anti-ceramide antibody before tape-stripping demonstrated an increased level of ASAH1b in the granular and spinous layers of the epidermis (Figure 5A) and a decreased level of ceramide (Figure 6A) in the SC of the depilatory cream-treated dorsal skin of TG mice compared to the depilatory cream-treated dorsal skin of WT mice. The increased level of ASAH1b and the decreased level of SC ceramide were corroborated by measurement with a Keyence Image Analyzer in the TG mice compared to the WT mice (Figure 5B and Figure 6B). Immunostaining with an anti-ceramide antibody after tape-stripping demonstrated that there was no further downregulation of the decreased level of ceramide (Figure 6A) in the SC of the depilatory cream-treated dorsal skin of TG mice. The significantly decreased level of SC ceramide in the dorsal skin of TG mice compared to WT mice even after tape-stripping was corroborated by measurement using a Keyence Image Analyzer (Figure 6B). Immunostaining with an anti-ASAH1b antibody after tape-stripping demonstrated that there was a further upregulation of the increased level of ASAH1b in the granular and spinous layers of the epidermis in the depilatory cream-treated dorsal skin of TG mice (Figure 5A). The significantly increased levels of ASAH1b after tape-stripping were corroborated by measurement using a Keyence Image Analyzer in the TG mice compared to the WT mice (Figure 5B). Under non-reduced conditions to prevent breaking the S–S bond between the β- and α-subunits of ASAH1, Western blotting revealed a distinct increase in ASAH1b in the epidermis of the depilatory cream-treated dorsal skin of TG mice compared to the depilatory cream-treated dorsal skin of WT mice before and after tape-stripping, although aggregates of ASAH1b proteins and other proteins appeared at the top of the gel due to the non-reduced conditions used (Figure 7).

### 2.6. Immunostaining of Macrophages Using Various Macrophage Markers, Mast Cells and IL-33 in the Dorsal Skin of TG and WT Mice before and after Tape-Stripping

Immunostaining with F4/80, CD80, and CD163 antibodies before tape-stripping showed that there was a significant increase in CD80^+^ M1 and CD163^+^ M2 macrophages in the depilatory cream-treated dorsal skin of TG mice compared to the depilatory cream-treated dorsal skin of WT mice, although the numbers of total macrophages did not differ between the WT and the TG mice (Figure 8A–C). Immunostaining with F4/80, CD80, and CD163 antibodies at day 2 after tape-stripping showed that there were slightly increased numbers of total M1 and M2 macrophages in the dermis of both the WT and TG mice, with the increased numbers of macrophages remaining unchanged in the TG mice compared to the WT mice (Figure 8A–C). Toluidine blue staining before tape-stripping revealed that there was a slight but not significant increase in the number of mast cells (Figure 9A). Toluidine blue staining after tape-stripping revealed that the number of mast cells was not altered in the dermis of the depilatory cream-treated WT or the depilatory cream-treated TG mice (Figure 9A). Immunostaining with an antibody to IL-33 before tape-stripping revealed that there was a slightly increased expression of IL-33 in the nuclei of keratinocytes in the spinous layer in the TG mice compared to the WT mice (Figure 9B). Immunostaining with antibodies to IL-33 after tape-stripping revealed that there was a further marked increase in the staining of IL-33 within the nuclei of keratinocytes in the spinous layer of the TG mice compared to the WT mice, although the number of IL-33^+^ cells remained unchanged with the TG mice still having a significantly higher number of IL-33^+^ cells than the WT mice (Figure 9B).

### 2.7. Effects of Overexpressing ASAH1b on the Activity of SGDase and ASAH1 in the Epidermis

Enzymatic analysis of epidermal homogenates revealed that SGDase activity was not significantly upregulated in the depilatory cream-treated dorsal skin of TG mice, which occurred at a negligible level both in the TG and in the WT mice before and at day 2 after tape-stripping (Figure 10). On the other hand, ASAH1 activity was detected at a similar level just above baseline both in the TG and in the WT mice before tape-stripping (Figure 11). This indicates that the overexpression of ASAH1b in the epidermis of TG mice does not elicit the enzymatic activation of SGDase and/or ASAH1, despite the fact that purified ASAH1b has the enzymatic activities of both SGDase and ASAH1 [22].

### 2.8. mRNA Expression Levels of Inflammatory Cytokines and Chemokines in the Skin of TG and WT Mice before and at Day 2 after Tape-Stripping

Real-time PCR analysis of skin samples before tape-stripping revealed that the mRNA expression levels of various cytokines and chemokines, including IL-5, CCL11, IL-22, CXCL10, and IFNγ, were significantly increased in the depilatory cream-treated dorsal skin of TG mice without any detectable skin inflammation, compared to the depilatory cream-treated dorsa skin of WT mice (Figure 12). On the other hand, the mRNA expression levels of CCL22 and TGFβ were significantly downregulated in the depilatory cream-treated dorsal skin of TG mice (Figure 12). At day 2 after the tape-stripping, the mRNA expression levels of several inflammatory cytokines and chemokines, including IL-4/IL-33/CXCL1/CXCL12/TLR9/CD163 and IL-5/IL-18/IL-22/CXCL10/IFNγ, were significantly up- and downregulated, respectively, in the TG mice (Figure 12). In contrast, in the depilatory cream-treated dorsal skin of WT mice, tape-stripping elicited a significant increase in caspase (Casp)3 only at the mRNA level and no cytokines or chemokines were downregulated at the mRNA level by tape-stripping (Figure 12).

## 3. Discussion

Recent evidence has shown that SGDase plays a crucial role in the ceramide deficiency of AD skin, which is an essential physiological factor for the disrupted barrier and water-holding functions of the SC that serve as a pathological trigger for the development of the recurrent dermatitis [7]. Further, SGDase has been recently identified as the β-subunit (ASAH1b) of ASAH1 [7,22]. Thus, we considered it important to determine whether overexpressing ASAH1b in the epidermis would cause the development of AD-like skin symptoms in mice.

In this study, when observed 3 days after hair removal using a depilatory cream containing glycolic acid, the 8-week-old TG mice without any visible skin inflammation had excoriated, scaly, and lichenized skin, as well as a mildly disrupted barrier function with significantly increased TEWL values, compared to the WT mice. The skin of the TG mice also had a decreased level of ceramide in the SC and an increased level of ASAH1b in the upper epidermis. Further, the increased level of ASAH1b protein in the epidermis of TG mice was confirmed by Western blotting. Although those findings suggest that the overexpression of ASAH1b in the epidermis might function as SGDase to reduce SC ceramide levels, which would result in the diminished barrier function, surprisingly, no SGDase enzyme activity could be detected in the epidermis of the TG or WT mice. On the other hand, the activity of ASAH1 occurred at a similar level in the TG and WT mice. Therefore, it is likely that the decreased level of SC ceramide observed in the TG mice is not attributable to the upregulated activity of either SGDase or ASAH1, which would reduce the levels of SC ceramide because the β-subunit of ASAH1 has a distinct enzyme activity of ASAH1 together with SGDase activity [22]. Although the reason(s) why SGDase activity was not detected in the epidermis of WT or TG mice remains unknown, it seems likely that additional biological factors and/or molecular modifications of the β-subunit are required for its enzymatic activation. Based on the appearance of the excoriated, scaly, and lichenized skin only in the depilatory cream-treated dorsal skin of TG mice, it seems reasonable to assume that the overexpression of ASAH1b in the epidermis, regardless of its enzymatic function, is mechanistically associated through an acquired high vulnerability to an irritant such as glycolic acid, with the excoriated, scaly, and lichenized skin as well as the mildly disrupted barrier function.

Our histological and immunohistochemical studies demonstrated that the skin of the TG mice has a thickened epidermis with mild hyperkeratosis, a marked accumulation of M1 and M2 macrophages around adipose tissue in the lower dermis and a marked increased expression of IL-33 in the nuclei of the keratinocytes in the spinous layer compared to the WT mice. All of those characteristics were exacerbated by tape-stripping together with an increased ASAH1b staining. To determine whether the overexpression of ASAH1b protein itself, regardless of its enzymatic activation, is a major pathogenic factor of AD, it was essential to associate the provoked cutaneous reactions in TG mice with the phenotypic characteristics of AD. As for the infiltration of macrophages, it has already been reported that the number of macrophages increases with tissue-specific heterogeneity both in acute and in chronically inflammatory AD skin [24]. It is also known that macrophages play key roles as antigen-presenting cells in the elicitation of AD by exhibiting different polarization with distinct functions categorized as M1 and M2 macrophages, depending on the acute or chronic status at the site of inflammation [25]. Since it has been established that tape-stripping tends to elicit macrophage infiltration [26], the infiltration of macrophages observed in AD skin [27,28] may reflect a high vulnerability of the skin to various stimuli. In the depilatory cream-treated dorsal skin of TG mice without any visible skin inflammation, there was a significantly increased number of M1 and M2 macrophages compared to the depilatory cream-treated dorsal skin of WT mice. That infiltration of macrophages was further upregulated by tape-stripping to a higher level in the skin of the TG mice than in the WT mice. This observation strongly suggests that the skin of the TG mice resembles the phenotypic characteristics of atopic dry skin [7] involved in AD lesions in which there is a decreased barrier function and a ceramide deficiency in the SC as well as macrophage infiltrations in the dermis [27,28], despite the lack of cutaneous inflammation. Consistently, the mRNA level of CD163 (a marker of M2 macrophages) was significantly increased after tape-stripping in the depilatory cream-treated dorsal skin of TG mice but not in the depilatory cream-treated dorsal skin of WT mice. In the mechanistic connection between macrophage accumulation and cytokine/chemokine profiles, M1 macrophages, which are activated by IFNγ, are predominantly associated with the expansion of inflammation and high antigen-presenting capabilities, by secreting proinflammatory cytokines (e.g., TNF-α, IL-6 and IL-12) and triggering Th1-polarized responses [29,30]. On the other hand, M2 macrophages, which are activated by IL-4 and IL-13, serve as anti-inflammatory and immunoregulatory factors. In the depilatory cream-treated dorsal skin of TG mice without any visible skin inflammation even before tape-stripping, the mRNA levels of several cytokines and chemokines, including IFNγ and IL-22, which activate macrophages [30] and are released by macrophages [31], respectively, are significantly upregulated compared to the depilatory cream-treated dorsal skin of WT mice. Further, mRNA levels of those factors, including IL-4, IL-33, and CD163, are significantly enhanced by tape-stripping. On the other hand, mRNA levels of IL-22 and IFNγ are significantly downregulated by tape-stripping. The expression profiles of cytokines and chemokines seem to reflect a certain skin integrity where the depilatory cream-treated dorsal skin of TG mice is predominantly situated under a condition that is highly sensitive to various environmental stimuli, which results in the infiltration of macrophages. It is probable that the macrophage–cytokine/chemokine network is partially activated in the depilatory cream-treated dorsal skin of TG mice without any visible skin inflammation because the same tape-stripping did not stimulate the mRNA levels of any of those factors, except for Casp3, in the depilatory cream-treated dorsal skin of WT mice.

As for the increased expression of IL-33 in the depilatory cream-treated dorsal skin of TG mice without any visible skin inflammation, available evidence indicates that intracellular IL-33, which is mainly distributed in the nuclei of basal and suprabasal epidermal keratinocytes, is distinctly expressed at higher levels throughout the epidermis of AD lesions compared with HC skin [32,33,34]. Our results show that IL-33 is expressed at a higher level in keratinocyte nuclei in the epidermis of the depilatory cream-treated dorsal skin of TG mice even without any visible skin inflammation compared to the depilatory cream-treated dorsal skin WT mice. Further, tape-stripping significantly upregulates the mRNA level of IL-33 only in the depilatory cream-treated dorsal skin of TG mice. Those findings strongly suggest that the skin integrity of the depilatory cream-treated dorsal skin of TG mice resembles atopic dry skin [7] with the skin signature of chronic dermatitis, and that the tape-stripped skin of the TG mice imitates cutaneous characterization, reflecting the non-allergic type of acute dermatitis.

Thus, it seems likely that the non-inflamed skin of the depilatory cream-treated TG mice occurs in a proinflammatory status including mild hyperkeratosis, conditions that could be ascribed to a mild barrier disruption that might occur due to the following speculated mechanism: The formation of functional lamellae requires that all components, proteins and lipids, are in the correct proportion and properly assembled. ASAH1 occurs in the released LGs outside of the cells and allows their self-assembly into the lamellae. In this case, the overexpression of ASAH1b stored in the LGs could result in skin functional abnormality as AD dry skin. Therefore, it is possible that such proinflammatory reactions in the depilatory cream-treated TG mice without any visible skin inflammation could be attributed to the hair removal process using the depilatory cream containing glycolic acid. If that is the case, it is likely that the depilatory cream-treated dorsal skin of TG mice acquires a higher susceptibility to environmental stimuli than the dorsal skin of the depilatory cream-treated WT mice. That skin sensitivity is consistent with our observations that even in ear skin not treated with a depilatory cream, the tape-stripping produced more severe dermatitis in TG mice than in WT mice, accompanied by a significantly increased ear thickness, compared to WT mice. Thus, these skin properties in the depilatory cream-treated TG mice resemble the AD diathesis that the “atopic skin” is generally associated with a lowered threshold of irritant responsiveness [28].

In conclusion, in this study, we clearly show that the overexpression of ASAH1b in the upper epidermis of the depilatory cream-treated TG mice causes the spontaneous development of skin reactions that resemble AD, although SGDase enzyme activity was not detectable in the epidermis of mice. These findings strongly indicate that the depilatory cream-treated dorsal skin of TG mice, even without any visible skin inflammation, is predominantly situated in an atopic dry skin condition that becomes highly sensitive to various environmental stimuli. The sum of our results suggests that ASAH1b itself, regardless of its enzymatic activity, is a major etiologic factor constituting atopic dry skin symptoms via an unknown mechanism, as might be speculated by the abnormal integrity of lipid lamellae inducible by an overly abundant ASAH1b.

## 4. Materials and Methods

### 4.1. Materials

Horseradish peroxidase-conjugated goat polyclonal anti-mouse IgG was obtained from Transduction Laboratories (Franklin Lakes, NJ, USA). β-actin was obtained from Cell Signaling Technology (Danvers, MA, USA). Sphingolipid ceramide N-deacylase from *Pseudomonas* sp. (Cat No. S2563-.25UN), and the antibody to ASAH1b and other reagent grade chemicals were purchased from Sigma Aldrich (Saint Louis, MO, USA).

### 4.2. Construction of Expression Vectors

The involucrin promoter-driven expression vector plasmid was a generous gift from Prof. Fiona Watt at Kings College London [35]. The WT ASAH1 (ASAH1 accession No. NM_019734) cDNA clone was obtained from RIKEN (Tsukuba, Japan) and was amplified by PCR using a primer located at the end of the signal peptide, 5′-GGCGCAGGTGACTGCCG-3′ as a reverse primer and a primer located at the start of the beta-subunit, 5′-TGTACATCAATCATAACTG-3′ as a forward primer. After amplification, the plasmid was obtained by self-ligation. After the sequence of ASAH1b was checked for the addition of the signal peptide sequence, it was amplified by PCR with the following primers: forward primer 5′-GCGGCCGCAAGATGCGGGGCCAAAGTCTTC-3′, reverse primer 5′-GCGGCCGCTCACCAGCCTATACAAGGGTC-3′. These primers had an added NotI site. The amplified PCR product was subcloned into the NotI site of the involucrin promoter-driven expression vector. The sequence of ASAH1b with the signal peptide was checked again to confirm the absence of nucleotide substitution and the direction of ASAH1b. The expression of ASAH1b protein was confirmed by transfecting the expression vector into HEK293T cells followed by Western blotting.

### 4.3. Production of ASAH1b-Overexpressing TG Mice

The expression vector was amplified and then digested with the restriction enzyme SalI to generate the transgene for injection. The transgene containing the involucrin promoter and the β-subunit of ASAH1 with the signal peptide was injected into 200 fertilized eggs of C57BL/6J mice, resulting in the generation of 2 F0 ASAH1b TG mice, which were mated and bred. The process of generating TG mice was outsourced to Uni-Tech Co. Ltd., Japan. All mice were kept in a SPF environment at the Experimental Animal Research Center of Jichi Medical University, with a constant temperature of 22–24 °C, a constant humidity of 60–75%, a 12 h dark/light cycle, with 4–5 mice per cage, free access to food and water, with regular changes of bedding. All mice were acclimated for 1–2 months and were then utilized for these experiments at 8 weeks of age. All animal care protocols and experiments were reviewed and approved by the Committee of the Laboratory Animal Research at Jichi Medical University and this study followed the ARRIVE animal experiment guidelines (https://arriveguidelines.org (accessed on 1 April 2021)).

### 4.4. Skin Eruption Scoring

Scaling, erythema, and edema were scored on a 1–5 grading basis (1 = none; 2 = slight; 3 = mild, 4 = moderate; 5 = severe) by a trained dermatologist (MK). The intensity of ear skin eruptions is expressed as total scores.

### 4.5. Measurement of Skin Barrier Function

Trans-epidermal water loss (TEWL), which is a parameter reflecting the barrier function of the skin, was measured using a Tewameter 300 (Courage and Khazaka GmbH, Cologne, Germany) 2 days after the dorsal skin of TG and WT mice was shaved with a depilatory cream (Reckitt Benckiser Japan Ltd., Tokyo, Japan) and is expressed as g/m^2^/h.

### 4.6. Tape-Stripping

One day after the dorsal or ear skin of TG and WT mice was shaved with a depilatory cream (Reckitt Benckiser Japan Ltd., Tokyo, Japan) or not, tape-stripping was repeated three times or five times, respectively, using adhesive tape (Nichiban, Tokyo, Japan). Two days after the tape-stripping, the skin was subjected to TEWL measurements and histological and immunological observations.

### 4.7. Histological Observation

Skin samples from the shaved dorsal or non-shaved ear skin of the TG and WT mice were fixed in formaldehyde solution and then embedded in paraffin. Paraffin blocks were sectioned into 4 to 6 μm thick specimens. Hematoxylin and eosin staining was performed as described previously [36]. Epidermal thickness was measured utilizing ImageJ software (imageJ 1.54g). Toluidine blue staining was performed as described previously [36]. Mast cells were observed using a Keyence BZ-700 microscope (Keyence, Tokyo, Japan), after which metachromatically stained mast cells were identified and counted by experienced researchers.

### 4.8. Immunohistochemical Staining

Skin specimens from shaved dorsal or non-shaved ear skins of TG and WT mice were fixed in formaldehyde and then embedded in paraffin blocks, which were thinly sliced at 5 μm. After deparaffinization and hydrophilization of the slides, antigen retrieval was performed in citrate buffer (pH 6.0) or EDTA buffer (pH 9.0) at 120 °C for 10 min. The tissues were then covered and incubated with blocking agent (Blocking One Histo; Nacalai Tesque Inc., Kyoto, Japan) for 30 min at room temperature. The slides were then incubated with primary antibodies as follows: anti-IL-33 antibody (Enzo Biochem, Inc., Framingdale, NY, USA, Nessy-1) [37], F4/80 antibody (BioRad, Richmond, CA, USA: MCA497G) [38], anti-ASAH1 β-subunit antibody (GeneTex, Irvine, CA, USA; GTX114267), anti-ceramide antibody (ENZO, New York, NY, USA; ALX-804-196-T050), anti-CD80 antibody (ProteinTech, Rosemont, IL, USA, B7-1) [39], and anti-CD163 antibody (ProteinTech, Rosemont, IL, USA; 16646-1-AP) [40]. After washing with PBS, the specimens were incubated with a biotin-labeled secondary antibody (ABC Kit; Vector Laboratories, Burlingame, CA, USA; PK-4001 or PK-4002) for 30 min, washed with PBS and incubated with an avidin–biotin complex solution (ABC Kit; Vector Laboratories, Burlingame CA, USA; PK-4001 or PK-4002) for 30 min. After washing with PBS, the slides were incubated with 3,3′-diaminobenzidine tetrahydrochloride (DAB; WAKO, Osaka, Japan) dissolved at 0.05% in 0.05 M Tris-HCl buffer with 0.03% H_2_O_2_ for 30 s to 2 min until the brown color was recognized as positive staining compared to the negative control. After washing with distilled water, the slides were further stained for nuclei by incubation with hematoxylin solution (Muto Pure Chemicals Co., Ltd., Tokyo, Japan). The immunohistochemical staining was observed using a Keyence BZ-700 microscope (Keyence, Tokyo, Japan), and the stained areas were measured using a Keyence Image Analyzer (Keyence, Tokyo, Japan).

### 4.9. Real-Time Polymerase Chain Reaction (RT-PCR)

The skin samples were homogenized and total RNAs were extracted utilizing a RNeasy Plus Universal Mini Kit (QIAGEN K.K., Tokyo, Japan). Each total RNA was reverse transcribed to a cDNA and was then subjected to RT-PCR analysis utilizing commercial primers as shown in Table 1.

### 4.10. Western Blotting Analysis

Biopsied skins were incubated overnight in 1% NaCl aqueous solution to separate the epidermis from the dermis. The isolated epidermis was homogenized for 30 s in a blender using T-PER Tissue Protein Extraction Reagent (Thermo Scientific, Waltham, MA, USA). The homogenates were centrifuged for 15 min at 12,000 rpm and the supernatants were passed through a Millex-HV filter (pore size = 0.45 μm), after which they were subjected to Western blot analysis to detect the levels of ASAH1b protein under non-reduced conditions (without 2-mercaptoethanol) to prevent breaking the S–S bond between the β- and α-subunits of ASAH1. Amounts of total protein were quantitated using the BCA protein reagent (Thermo Scientific, Rockford, IL, USA). Total proteins (10~20 μg/lane) without denaturation by heating were separated by electrophoresis on 12% sodium dodecyl sulfate (SDS)–polyacrylamide gels (BioRad, Richmond, CA, USA). After electrophoresis, proteins were transferred onto polyvinylidene difluoride (PVDF) membranes and were immunoblotted with the anti-ASAH1b antibody (GeneTex, Irvine, CA, USA). Detection was performed using donkey anti-goat IgG H&L, anti-rabbit IgG-horseradish peroxidase-linked secondary antibody (Cell Signaling, Danvers, MA, USA). A goat polyclonal antibody to β-actin (Abcam, Cambridge, UK) was used as an internal control. Proteins were visualized using an enhanced chemiluminescence kit (GE Healthcare, Little Chalfont, UK). Images were taken using ImageQuant LAS 4000 Version 1.0 (GE Healthcare).

### 4.11. Assays for SM Deacylase and ASAH1 Activities

The epidermis of WT and TG mice at the age of 8 weeks was separated by incubating the skin overnight in 1 M NaCl aqueous solution and was homogenized in acetic acid buffer solution. The epidermal homogenate was treated with T-PER™ Tissue Protein Extraction Reagent (Thermo Scientific, Rockford, IL, USA, Cat No. 78510) and centrifuged at 12,000 rpm for 15 min. The supernatants were passed through a 0.45 μm filter (Millex-HV, Merck, Darmstadt, Germany) to obtain the enzyme solutions. The enzymatic assays for SGDase and ASAH1 were carried out using the enzyme solutions according to modified methods as described previously [22]. For the SGDase assay, the enzyme solution was incubated for 6 h at 37 °C with SM (1 nmoL/200 μL) in 150 μL 50 mM sodium acetate buffer solution (pH 4.7) with 20 mM CaCl_2_. The reactions were terminated by the addition of two volumes of chloroform/methanol (2:1, *v*/*v*). The reaction solutions were filtered through a 0.45 µm filter (Millex-HV, Merck, Darmstadt, Germany) to obtain separated oil and water layers. The water layer was subjected to LC/MS/MS analysis for SPC by the standard addition method. For the ASAH1 assay, the enzyme solution (adjusted to 400 ng protein/μL) was incubated for 6 h at 37 °C with C18-ceramide (2 nmoL/200 μL) in 150 μL 50 mM sodium acetate buffer solution (pH 4.7). The reactions were terminated by the addition of two volumes of chloroform/methanol (2:1, *v*/*v*). The reaction solutions were passed through a 0.45 µm filter (Millex-HV, Merck, Darmstadt, Germany) to obtain separated oil and water layers. The oil layer was subjected to LC/MS/MS analysis for sphingosine by the standard addition method.

### 4.12. Statistical Analysis

All data are expressed as means ± SD (n = 3~9) unless noted otherwise. For pairwise comparisons, Student’s *t*-test was used. For multiple comparisons, data were tested using Tukey’s multiple comparison test or Mann–Whitney test. *p* values less than 0.05 are considered statistically significant.

## Figures and Tables

**Figure 1 ijms-25-08737-f001:**
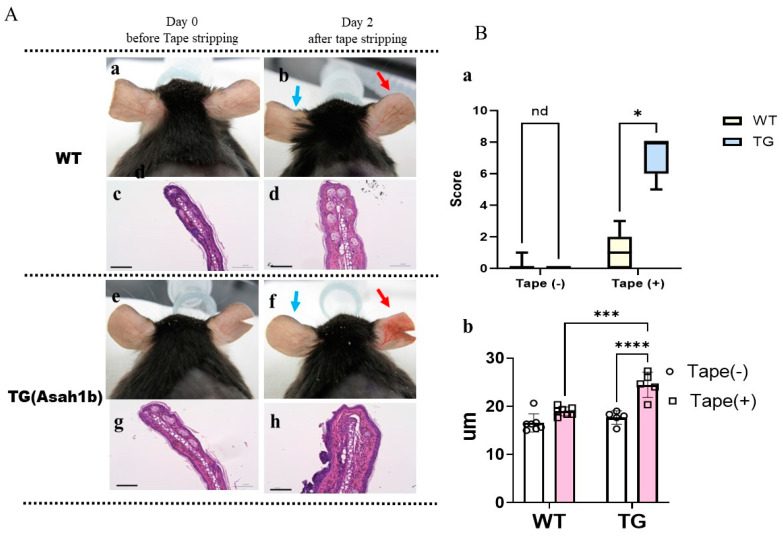
Appearance of the ear skin, H&E staining, skin score and thickness of the epidermis before and after repeated tape-stripping. (**A**) (**a**) WT ears at day 0 before tape-stripping, (**b**) WT ears at day 2 after tape-stripping of the right ear (red arrow); the left ear (blue arrow) was not tape-stripped as a control, (**c**) H&E staining of the left WT ear at day 2, (**d**) H&E staining of right WT ear at day 2 after tape-stripping, (**e**) TG ears at day 0 before tape-stripping, (**f**) TG ears at day 2 after tape-stripping of the right ear (red arrow); the left ear (blue arrow) was not tape-stripped as a control, (**g**) H&E staining of the left TG ear at day 2, (**h**) H&E staining of the right TG ear at day 2 after tape-stripping. (**B**) (**a**) Total visible score on the outside ear skin of TG and WT mice at the age of 8 weeks before and day 2 after tape-stripping. n = 7 for WT, n = 5 for TG, *: *p* < 0.05 by Mann–Whitney comparisons test. (**b**) Thickness of the epidermis of the outside of the ear measured using Image J software (imageJ 1.54g). n = 5~7, ***: *p* < 0.001, ****: *p* < 0.0001 by Tukey’s multiple comparisons test. nd: not significant difference.

**Figure 2 ijms-25-08737-f002:**
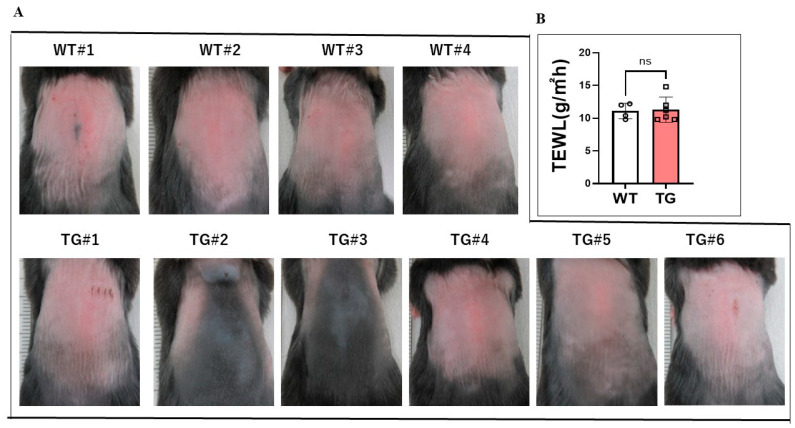
Surface appearance of the dorsal skin of TG and WT mice at the age of 8 weeks after hair removal using electric hair clippers and barrier function values three days after hair removal. (**A**) Skin appearance three days after hair removal. (**B**) Barrier function measured by TEWL three days after hair removal, TG; n = 6, WT; n = 4, ns: not significant, by Mann–Whitney test.

**Figure 3 ijms-25-08737-f003:**
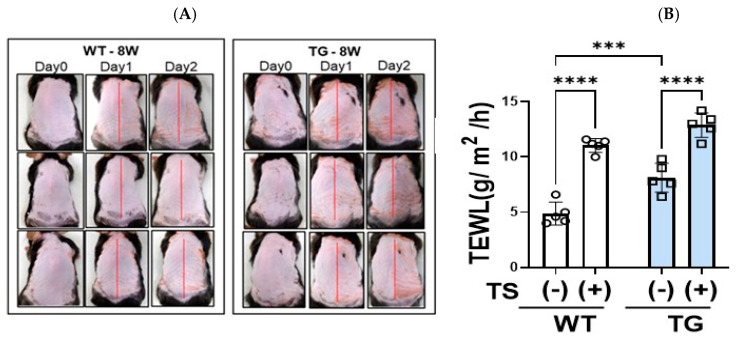
Surface appearance of the depilatory cream-treated dorsal skin of TG and WT mice at the age of 8 weeks and barrier function before and at day 2 after tape-stripping. (**A**) Skin appearance at Days 0, 1 and 2, **right** and **left** sides (separated by red line) of the dorsal skin show tape-stripped and non-tape-stripped areas, respectively. (**B**) Barrier function measured by TEWL before and at day 2 after tape-stripping, n = 5, ****: *p* < 0.0001, ***: *p* < 0.001 by Tukey’s multiple comparisons test.

**Figure 4 ijms-25-08737-f004:**
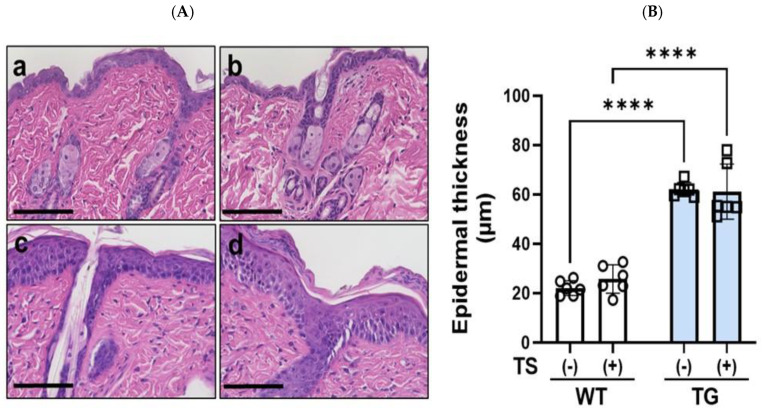
H&E staining of the depilatory cream-treated dorsal skin of TG and WT mice at the age of 8 weeks before and after tape-stripping. (**A**) H&E staining, (**a**) WT dorsal skin before tape-stripping, (**b**) WT dorsal skin at day 2 after tape-stripping, (**c**) TG dorsal skin before tape-stripping, (**d**) TG dorsal skin at day 2 after tape-stripping, bars = 100 μm, (**B**) Epidermal thickness, analysis by measurement using Image J software (imageJ 1.54g). n = 6, ****: *p* < 0.0001 by Tukey’s multiple comparisons test.

**Figure 5 ijms-25-08737-f005:**
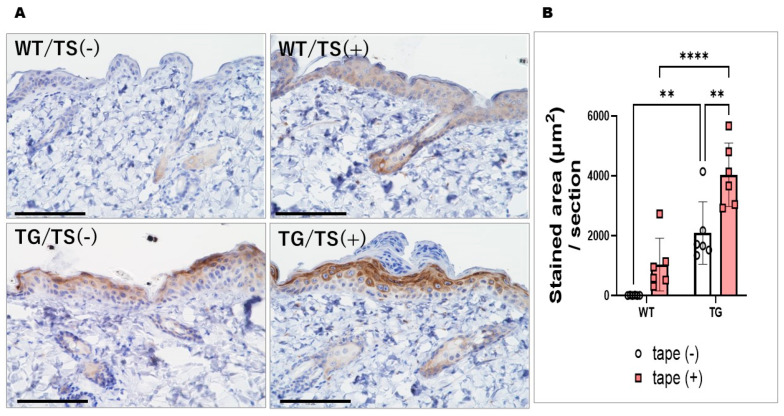
Immunostaining of ASAH1 with an anti-ASAH1b antibody in the depilatory cream-treated dorsal skin of TG and WT mice at the age of 8 weeks before and at day 2 after tape-stripping. (**A**) Immunostaining, bars = 100 μm, (**B**) Stained area, analysis by measurement with a Keyence Image Analyzer. n = 6, ****: *p* < 0.0001, **: *p* < 0.01 by Tukey’s multiple comparisons test. TS = tape-stripped.

**Figure 6 ijms-25-08737-f006:**
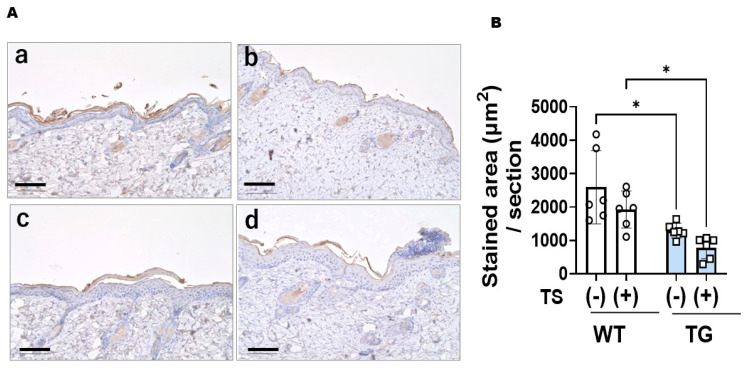
Immunostaining of ceramide with an anti-ceramide antibody in the depilatory cream-treated dorsal skin of TG and WT mice at the age of 8 weeks before and at day 2 after tape-stripping. (**A**) Immunostaining, bars = 100 μm, (**a**) WT/TS (−), (**b**) WT/TS (+), (**c**) TG/TS (−), (**d**) TG/TS (+), (**B**) Stained area, analysis by measurement with a Keyence Image Analyzer, n = 6, *: *p* < 0.05 by Tukey’s multiple comparisons test. TS: tape-stripped.

**Figure 7 ijms-25-08737-f007:**
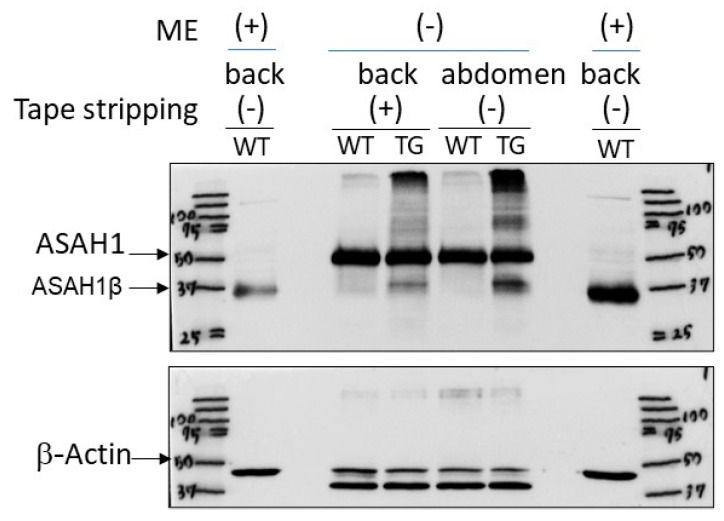
Western blotting of ASAH1b protein in epidermal homogenates of the depilatory cream-treated dorsal skin and abdominal skin of TG and WT mice at the age of 8 weeks before and at day 2 after tape-stripping. ME = treated with 2-mercaptoethanol. The epidermal homogenates were obtained at 8 weeks of age or at day 2 after tape-stripping.

**Figure 8 ijms-25-08737-f008:**
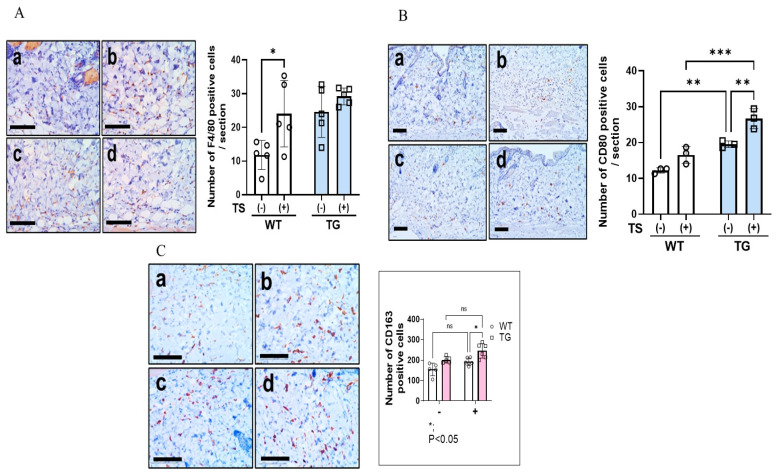
Immunostaining of macrophages using various macrophage markers in the depilatory cream-treated dorsal skin of TG and WT mice at the age of 8 weeks before and at day 2 after tape-stripping. (**A**) Immunostaining of F4/80, bars = 100 μm, analysis by measurement with a Keyence Image Analyzer, n = 5, *: *p* < 0.05 by Tukey’s multiple comparisons test. (**B**) Immunostaining of CD80, bars = 100 μm, analysis by measurement with a Keyence Image Analyzer, n = 3, ***: *p* < 0.001, **: *p* < 0.01 by Tukey’s multiple comparisons test. (**C**) Immunostaining of CD163, bars = 100 μm, analysis by measurement with a Keyence Image Analyzer, n = 5, *: *p* < 0.05, by Tukey’s multiple comparisons test. (**a**) WT/TS (−), (**b**) WT/TS (+), (**c**) TG/TS (−), (**d**) TG/TS (+), TS: Tape-stripped. ns: not significant.

**Figure 9 ijms-25-08737-f009:**
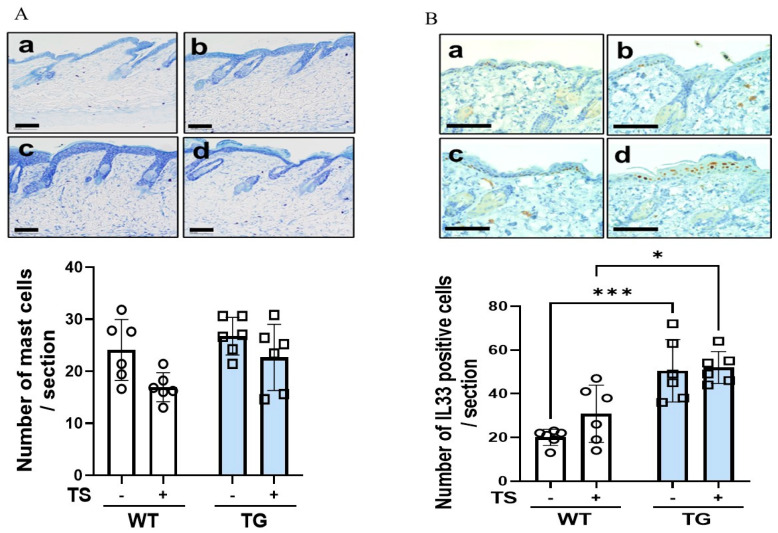
Toluidine blue staining of mast cells (**A**) and immunostaining with an anti-IL-33 antibody (**B**) in the depilatory cream-treated dorsal skin of TG and WT mice at the age of 8 weeks before and at day 2 after tape-stripping. bars = 100 μm, analysis by measurement with a Keyence Image Analyzer. n = 6, ***: *p* < 0.001, *: *p* < 0.05 by Tukey’s multiple comparisons test. (**a**) WT/TS (−), (**b**) WT/TS (+), (**c**) TG/TS (−), (**d**) TG/TS (+), TS: Tape-stripping.

**Figure 10 ijms-25-08737-f010:**
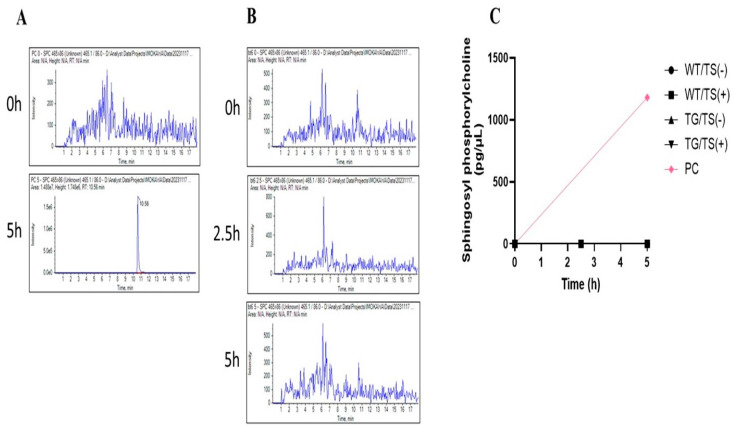
Effects of the overexpression of ASAH1b on the activity of SGDase in the epidermis. (**A**): LC-MS-MS spectrum for released SPC by sphingolipid ceramide N-deacylase from *Pseudomonas* sp. as a positive control. (**B**): LC-MS-MS spectrum for released SPC by SGDase in the epidermal homogenate from TG/TS (+). (**C**): Time course of the released SPC after enzymatic reaction with epidermal homogenate or sphingolipid ceramide N-deacylase from *Pseudomonas* sp. as positive control.An epidermal homogenate was incubated for 5 h at 37 °C with varying amounts of SM. The enzymatic activities expressed as released SPC for WT/TS(-), WT/TS(+), TG/TS(-), and TG/TS(+) were located at O level. The final reaction mixtures contained 50 mM potassium acetate buffer (pH 4.7), the enzyme source, the substrate, 0.1% Triton X-100 and 20 mM CaCl_2_. The rate of SPC generation was measured as a function of SGDase by LC-MS-MS analysis. PC: positive control using sphingolipid ceramide N-deacylase from *Pseudomonas* sp. as SGDase. TS: Tape-stripping.

**Figure 11 ijms-25-08737-f011:**
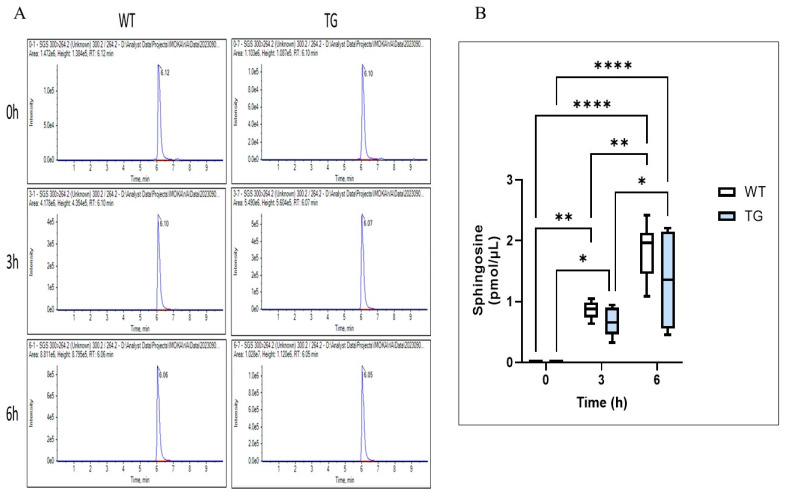
Effects of the overexpression of ASAH1b on the activity of ASAH1 in the epidermis. (**A**): LC-MS-MS spectrum of released sphingosine (SPH) after enzymatic reaction with epidermal homogenate. (**B**): Enzymatic activity of ASAH1 expressed as released SPH in epidermal homogenates of WT and TG mice. An epidermal homogenate was incubated for 12 h at 37 °C with varying amounts of ceramide substrates. The final reaction mixtures contained 50 mM potassium acetate buffer (pH 4.7), the enzyme source, the substrate, 0.1% Triton X-100 and 20 mM CaCl_2_. Blue rectangles indicate TG mice. The rate of sphingosine (SPH) generation was measured as a function of ASAH1 by LC-MS-MS analysis. n = 6, ****: *p* < 0.0001, **: *p* < 0.01, *: *p* < 0.05 by Tukey’s multiple comparisons test.

**Figure 12 ijms-25-08737-f012:**
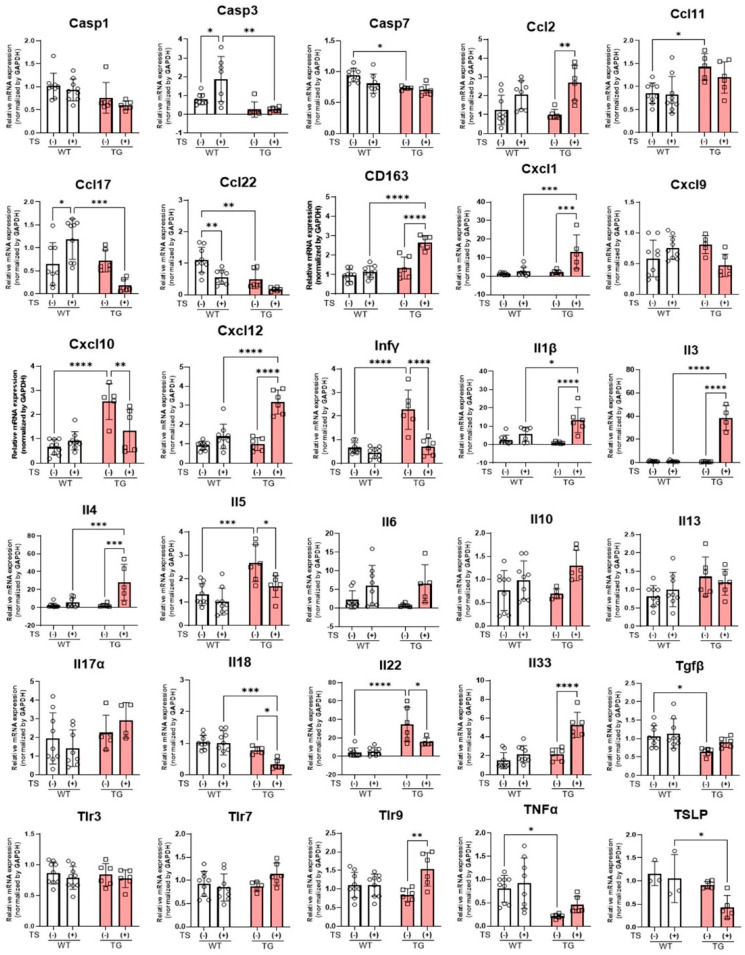
mRNA expression levels of inflammatory cytokines and chemokines in the depilatory cream-treated dorsal skin of TG and WT mice at the age of 8 weeks before and at day 2 after tape-stripping. n = 3~9, ****: *p* < 0.0001, ***: *p* < 0.001, **: *p* < 0.01, *: *p* < 0.05 by Tukey’s multiple comparisons test.

**Table 1 ijms-25-08737-t001:** Primer sequences of cytokines and chemokines used for RT-PCR analysis.

Symbol	Gene Name		Sequence (5′ > 3′)
Casp1	caspase 1	F	TGCCTGGTCTTGTGACTTGGA
		R	CCTATCAGCAGTGGGCATCTGTA
Casp3	caspase 3	F	AGCCATGGGCACATCTTCAG
		R	TGGTAACTTGGACATCATCCACAC
Casp7	caspase 7	F	AAGCCACTGCCTGAGATGGAA
		R	GGAATTATAAGGCGCTGGTGGA
Ccl2	chemokine (C-C motif) ligand 2	F	AGCAGCAGGTGTCCCAAAGA
		R	GTGCTGAAGACCTTAGGGCAGA
Ccl11	chemokine (C-C motif) ligand 11	F	CAGATGCACCCTGAAAGCCATA
		R	TGCTTTGTGGCATCCTGGAC
Ccl17	chemokine (C-C motif) ligand 17	F	CCGAGAGTGCTGCCTGGATTA
		R	AGCTTGCCCTGGACAGTCAGA
CCL22	chemokine (C-C motif) ligand 22	F	CTGACGAGGACACATAACATCATGG
		R	CTTCACTAAACGTGATGGCAGAGG
CD163	CD163 antigen	F	GCCAAACCGTGGAGTCACAG
		R	GGACCAATAGAATGGCTCCACAA
Cxcl1	chemokine (C-X-C motif) ligand 1	F	TGCACCCAAACCGAAGTC
		R	GTCAGAAGCCAGCGTTCACC
Cxcl9	chemokine (C-X-C motif) ligand 9	F	GAAGTGTGGACAGGGCCAAGTTA
		R	GACGGAACTTCTGCCCAGAGAC
Cxcl10	chemokine (C-X-C motif) ligand 10	F	ATCCGGAATCTAAGACCATCAAGAA
		R	GGACTAGCCATCCACTGGGTAAAG
CXCL12	chemokine (C-X-C motif) ligand 12	F	CAGAGCCAACGTCAAGCATC
		R	TTAATTTCGGGTCAATGCACAC
GAPDH	glyceraldehyde-3-phosphate dehydrogenase	F	TGTGTCCGTCGTGGATCTCTGA
		R	TTGCTGTTGAAGTCGCAGGAG
Ifng	interferon gamma	F	CGGCACAGTCATTGAAAGCCTA
		R	GTTGCTGATGGCCTGATTGTC
Il1b	interleukin 1 beta	F	TCCAGGATGAGGACATGAGCAC
		R	GAACGTCACACACCAGCAGGTTA
Il3	interleukin 3	F	ACCGTTTAACCAGAACGTTGAATTG
		R	TCCACGAATTTGGACAGGTTTACTC
Il4	interleukin 4	F	TCTCGAATGTACCAGGAGCCATATC
		R	AGCACCTTGGAAGCCCTACAGA
Il5	interleukin 5	F	TCAGCTGTGTCTGGGCCACT
		R	TTATGAGTAGGGACAGGAAGCCTCA
Il6	interleukin 6	F	CCACTTCACAAGTCGGAGGCTTA
		R	CCAGTTTGGTAGCATCCATCATTTC
Il10	interleukin 10	F	GCCAGAGCCACATGCTCCTA
		R	GATAAGGCTTGGCAACCCAAGTAA
Il13	interleukin 13	F	CAATTGCAATGCCATCTACAGGAC
		R	CGAAACAGTTGCTTTGTGTAGCTGA
Il17α	interleukin 17 alpha	F	ACGCGCAAACATGAGTCCAG
		R	AGGCTCAGCAGCAGCAACAG
Il18	interleukin 18	F	AAGACTCTTGCGTCAACTTCAAGGA
		R	ATGCGGCCAAAGTTGTCTGATTC
Il22	interleukin 22	F	ACATCAGCCGTGACGACCAG
		R	CGCTCAGACGCAAGCATTTC
Il33	interleukin 33	F	GATGAGATGTCTCGGCTGCTTG
		R	AGCCGTTACGGATATGGTGGTC
Tgfb1	transforming growth factor, beta 1	F	GTGTGGAGCAACATGTGGAACTCTA
		R	CGCTGAATCGAAAGCCCTGTA
TLR3	toll-like receptor 3	F	AAATCCTTGCGTTGCGAAGTG
		R	TCAGTTGGGCGTTGTTCAAGA
TLR7	toll-like receptor 7	F	CTTTGCAACTGTGATGCTGTGTG
		R	ACCTTTGTGTGCTCCTGGACCTA
TLR9	toll-like receptor 9	F	GAGACCCTGGTGTGGAACATC
		R	ACTGCAGCCTGTACCAGGAG
TNFa	tumor necrosis factor	F	TATGGCCCAGACCCTCACA
		R	GGAGTAGACAAGGTACAACCCATC
TSLP	thymic stromal lymphopoietin	F	AATGACCACTGCCCAGGCTA
		R	TTGTGAGGTTTGATTCAGACAGATG

## Data Availability

Data are contained within the article.

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
