# Peer review of "Overexpression of the β-Subunit of Acid Ceramidase in the Epidermis of Mice Provokes Atopic Dermatitis-like Skin Symptoms"

_ijms, 2024, doi:10.3390/ijms25168737_

Round 1

Reviewer 1 Report

Comments and Suggestions for Authors

The manuscript by Sashikawa-Kimura et al. entitled “Overexpression of the betta-subunit of acid ceramidase in the epidermis of mice provokes atopic dermatitis-like skin symptoms” provides very important and intriguing observation regarding potential involvement of ASAH1b subunit in the development of eczema phenotype (atopic dermatitis, AD). Presented work suggests that, when ASAH1b is overexpressed in mouse skin, this leads to an excessive sensitization of the skin to external insults and AD-like phenotype regardless there are no changes in measured ASAH1b or acid ceramidase (aCERase) enzymatic activity. The authors also demonstrated by means of immunohistochemistry that overexpression of ASAH1b results in the decrease in total ceramide level (Fig. 5). This brings to a conclusion that ASAH1b affects (decreases) ceramide level by yet to identify mechanism but not through the known enzymatic activities. Same is applied to the role of ASAH1b in the development of AD-type phenotype.

To the opinion of the review, the manuscript is missing two important parameters related to overexpression of ASAH1b in mouse skin.

1.    While successful overexpression of ASAH1b is demonstrated by immunohistochemistry and Western blotting (Fig. 4 and Fig. 6), there is no indication to what happens with the expression of ASAH1a subunit. If ASAH1a subunit expression is somehow linked to the expression of ASAH1b subunit, then one could expect the increase in functioning ASAH1 (acid ceramidase, aCDase) enzyme. Yet, there is no increase in aCDase activity. Is it due to improper linking of ASAH1a and ASAH1b subunits? Bottom line, it is important to know what happens to the expression of ASAH1a subunit.

2.    The lack of the increase in aCDase activity upon overexpression of ASAH1b subunit but the decrease in ceramides observed histochemically suggests the existence of other mechanism(s) leading the decrease in skin ceramide level. To this end, extensive sphingolipidomic analysis by the LC-MS/MS is needed to understand what happens to the actual levels of free sphingoid bases, ceramides, and sphingomyelins in the skin of transgenic animals versus wild-type mice.

Author Response

To the Reviewer 1

Thank you very much for your invaluable comments and suggestions.

We ‘d like to answer to your comments and suggestions in step by step manner as follows;

Reviewer 1

The manuscript by Sashikawa-Kimura et al. entitled “Overexpression of the betta-subunit of acid ceramidase in the epidermis of mice provokes atopic dermatitis-like skin symptoms” provides very important and intriguing observation regarding potential involvement of ASAH1b subunit in the development of eczema phenotype (atopic dermatitis, AD). Presented work suggests that, when ASAH1b is overexpressed in mouse skin, this leads to an excessive sensitization of the skin to external insults and AD-like phenotype regardless there are no changes in measured ASAH1b or acid ceramidase (aCERase) enzymatic activity. The authors also demonstrated by means of immunohistochemistry that overexpression of ASAH1b results in the decrease in total ceramide level (Fig. 5). This brings to a conclusion that ASAH1b affects (decreases) ceramide level by yet to identify mechanism but not through the known enzymatic activities. Same is applied to the role of ASAH1b in the development of AD-type phenotype.

To the opinion of the review, the manuscript is missing two important parameters related to overexpression of ASAH1b in mouse skin.

  1. While successful overexpression of ASAH1b is demonstrated by immunohistochemistry and Western blotting (Fig. 4 and Fig. 6), there is no indication to what happens with the expression of ASAH1a subunit. If ASAH1a subunit expression is somehow linked to the expression of ASAH1b subunit, then one could expect the increase in functioning ASAH1 (acid ceramidase, aCDase) enzyme. Yet, there is no increase in aCDase activity. Is it due to improper linking of ASAH1a and ASAH1b subunits? Bottom line, it is important to know what happens to the expression of ASAH1a subunit.

Response;

In the transgenic mice, we overexpressed only ASAH1b which is autocleavage unit from full ASAH1, leaving ASAH1a. Because ASAH1a subunit expression is not linked to the expression of ASAH1b subunit in this study, it is not important to look at what happens with the expression of ASAH1a subunit.  In addition, a good antibody to ASAH1a subunit is not commercially available.

------------------------------------------------------------------------------------------

  1. The lack of the increase in aCDase activity upon overexpression of ASAH1b subunit but the decrease in ceramides observed histochemically suggests the existence of other mechanism(s) leading the decrease in skin ceramide level. To this end, extensive sphingolipidomic analysis by the LC-MS/MS is needed to understand what happens to the actual levels of free sphingoid bases, ceramides, and sphingomyelins in the skin of transgenic animals versus wild-type mice.

Response

In the transgenic mice with overexpression of ASAH1b subunit, due to the lack of the increase of aCDase activity as well as of SMDase activity, the decreased level of ceramide in the stratum corneum and barrier disruption are very weak. So, we could not consider barrier dysfunction as a main causative factor for enhanced sensitivity to stimuli. We’d like to consider biological properties of ASAH1b with many cysteine residues like chemokines as a causative factor for enhanced sensitivity to stimuli regardless of its enzymatic activity. Of course, if aCDase activity and/or SMDase activity are up-regulated, the latter of which occurs in AD skin, more marked decrease of ceramide in the stratum corneum result in eliciting more marked barrier disruption, leading to more enhanced sensitivity to stimuli as frequently observed in AD skin.

Therefore, quantitative analysis of ceramide contents and other ceramide-related factors by the LC-MS/MS is not so important in this study and would not provide a further support for our conclusion.

The related sentences were added at the end of Discussion section. Corrected, modified or added sentences were shown by blue highlight in the revised manuscript with remarks.

-------------------------------------------------------------------------------------------------

Reviewer 2 Report

Comments and Suggestions for Authors

The authors produced a transgenic mice expressing high level of ASAH1b/SGDase in the upper epidermis. The main conclusion is that the skin of the transgenic mice presents a lowered threshold of sensitivity to environmental stimuli (hair removal using a depilatory cream or tape stripping) in comparison to the skin of wild type mice. This phenotype is reminiscent of some aspects of the atopic dry skin phenotype. These observations are based on macroscopic and histological observation of the mice skin, analysis of the inflammatory response by immunostaining and quantitative RT-PCR, and analysis of the activity of enzymes involved in SC ceramide metabolism.

The data presented by the authors might interest scientists working in the field of skin ceramides and their role in barrier function. However, the article needs to be thoroughly revised, both in content and form.

Some major points:

1/ The decrease in ceramide quantity in the SC is assessed only by immunohistochemistry, a global and semi-quantitative method. A more sensitive and precise quantitative method, such as LC-MS/MS, should be used. This is important because it is a major element in characterizing the phenotype of the transgenic mice.

2/ In a previous work, the authors reported that SGDase activity was increased in the skin of patients with AD (Imokawa et al., Int J Mol Sci 2021,22, (4)). Surprisingly, however, the overexpression of SGDase in the skin of mice does not induce any increase in the activity of this enzyme. This is another major point that needs to be addressed and discussed by the authors.

3/ The enzyme activity assays (results, paragraph 2.6) are incomprehensible. The incubation time is 12h at 37°C, thus the graphs present data from the first 5 or 6h of incubation? For aCDase activity, the authors state that there is a “similar distinct level both in TG and WT”, what does it mean?

4/ The transgene expression could have an impact in the regulation of the ceramide biosynthesis pathway in the skin of the transgenic mice. In addition, the expression of enzymes involved in this lipid biosynthesis pathway could be perturbed in the skin of patients with AD. It would thus be interesting to analyse the expression of such enzymes (like stearoyl CoA desaturase, elongases, ceramide synthases etc.) in the transgenic mice in comparison to the wild type mice.

5/ The article is too long, dense, and lacks fluidity. The sentences are often too long. There are unnecessary repetitions (e.g., at the age of 8 weeks, without any visible skin inflammation).

6/ The logic of the structure proposed by the authors is unclear. For example, in the results section, the authors could first present the analysis of transgene expression (+ enzyme activity, SC ceramide levels) corresponding to the current paragraph 2.4 (whose title is the same as 2.3!), then the macroscopic and histological study of the phenotype (paragraphs 2.1, 2.2, and 2.3 grouped together), and finally the analysis of the inflammatory phenotype (paragraphs 2.5 and 2.7 grouped together).

7/ The use of enzyme abbreviations should be standardized (e.g., ASAH1b/SGDase) and the conventional nomenclature should be used (e.g., BGCase versus GBA; aCDase versus ASAH). This is important especially for readers who are not familiar with the SC lipid biosynthesis and metabolic pathways.

Comments on the Quality of English Language

The English is understandable, but it is mainly the form that needs improvement.

Avoid repetitions throughout the text (e.g. "at the age of 8 weeks", "without any visible skin inflammation").

Shorter sentences would greatly facilitate understanding.

Author Response

To the Reviewer 2

Thank you very much for your invaluable comments and suggestions.

We ‘d like to answer to your comments and suggestions in step by step manner as follows;

Reviewer 2

The authors produced a transgenic mice expressing high level of ASAH1b/SGDase in the upper epidermis. The main conclusion is that the skin of the transgenic mice presents a lowered threshold of sensitivity to environmental stimuli (hair removal using a depilatory cream or tape stripping) in comparison to the skin of wild type mice. This phenotype is reminiscent of some aspects of the atopic dry skin phenotype. These observations are based on macroscopic and histological observation of the mice skin, analysis of the inflammatory response by immunostaining and quantitative RT-PCR, and analysis of the activity of enzymes involved in SC ceramide metabolism.

The data presented by the authors might interest scientists working in the field of skin ceramides and their role in barrier function. However, the article needs to be thoroughly revised, both in content and form.

Some major points:

1/ The decrease in ceramide quantity in the SC is assessed only by immunohistochemistry, a global and semi-quantitative method. A more sensitive and precise quantitative method, such as LC-MS/MS, should be used. This is important because it is a major element in characterizing the phenotype of the transgenic mice.

Response;

In the transgenic mice with overexpression of ASAH1b subunit, due to the lack of the increase of aCDase activity as well as of SMDase activity, the decrease of ceramide in the stratum corneum and barrier disruption are very weak. So, we could not consider barrier dysfunction as a causative factor for enhanced sensitivity to stimuli. We’d like to consider biological properties of ASAH1b with many cysteine residues like chemokines as a causative factor for enhanced sensitivity to stimuli regardless of its enzymatic activity. Of course, if aCDase activity and/or SMDase activity are up-regulated, the latter of which occurs in AD skin, more marked decrease of ceramide in the stratum corneum result in eliciting more marked barrier disruption, leading to more enhanced sensitivity to stimuli as observed in AD skin.

Therefore, quantitative analysis of ceramide contents and ceramide-related factors by the LC-MS/MS is not so important in this study and would not provide a further support for our conclusion.

Corrected, modified or added sentences were shown by blue highlights in the revised manuscript with remarks. The related sentences were added at the end of Discussion section.

---------------------------------------------------------------------------------------------------

2/ In a previous work, the authors reported that SGDase activity was increased in the skin of patients with AD (Imokawa et al., Int J Mol Sci 2021,22, (4)). Surprisingly, however, the overexpression of SGDase in the skin of mice does not induce any increase in the activity of this enzyme. This is another major point that needs to be addressed and discussed by the authors.

Response:

We have added some reasons why the overexpression of SGDase in the skin of mice does not induce any increase in the activity of this enzyme as described in Discussion section. Corrected, modified or added sentences were shown by blue highlights in the revised manuscript with remarks.

------------------------------------------------------------------------------------

3/ The enzyme activity assays (results, paragraph 2.6) are incomprehensible. The incubation time is 12h at 37°C, thus the graphs present data from the first 5 or 6h of incubation? For aCDase activity, the authors state that there is a “similar distinct level both in TG and WT”, what does it mean?

Response:

We are sorry to make a mistake for 12h. 5 h is correct. So, we have replaced 12 h with 5 h in the corresponding text. Further, according to the reviewer’s comments, we have modified the corresponding sentences in Results, paragraph 2.6 as described in the revised manuscript to make them clearer. Corrected, modified or added sentences were shown by blue highlight in the revised manuscript with remarks.

-------------------------------------------------

4/ The transgene expression could have an impact in the regulation of the ceramide biosynthesis pathway in the skin of the transgenic mice. In addition, the expression of enzymes involved in this lipid biosynthesis pathway could be perturbed in the skin of patients with AD. It would thus be interesting to analyse the expression of such enzymes (like stearoyl CoA desaturase, elongases, ceramide synthases etc.) in the transgenic mice in comparison to the wild type mice.

Response:

It seems unlikely that the transgene expression gives an impact in the regulation of the ceramide biosynthesis pathway because of the lack of SMDase activity as well as of unchanged acid ceramidase activity. As mentioned in the response to No 1 question, we could not consider barrier dysfunction as a causative factor for enhanced sensitivity to stimuli. We’d like to consider biological properties of ASAH1b with many cysteine residues like chemokines as a causative factor for enhanced sensitivity to stimuli, regardless of its enzymatic activity. So, analysis of several ceramide-biosynthesis enzymes would not provide a further support for our conclusion.

-----------------------------------------------------------------------

5/ The article is too long, dense, and lacks fluidity. The sentences are often too long. There are unnecessary repetitions (e.g., at the age of 8 weeks, without any visible skin inflammation).

Response:

According to the reviewer’s comments, we have modified corresponding sentences by deleting unnecessary repetitions, the corrected parts of which were shown by blue highlights throughout the revised manuscript with remarks.

------------------------------------------------------------------------------^^

6/ The logic of the structure proposed by the authors is unclear. For example, in the results section, the authors could first present the analysis of transgene expression (+ enzyme activity, SC ceramide levels) corresponding to the current paragraph 2.4 (whose title is the same as 2.3!), then the macroscopic and histological study of the phenotype (paragraphs 2.1, 2.2, and 2.3 grouped together), and finally the analysis of the inflammatory phenotype (paragraphs 2.5 and 2.7 grouped together).

Response:

According to the reviewer’s comments, we have modified corresponding sentences.

Corrected, modified or added sentences were shown by blue highlight in the revised manuscript with remarks.

---------------------------------------------------------------

7/ The use of enzyme abbreviations should be standardized (e.g., ASAH1b/SGDase) and the conventional nomenclature should be used (e.g., BGCase versus GBA; aCDase versus ASAH). This is important especially for readers who are not familiar with the SC lipid biosynthesis and metabolic pathways.

Response:

According to the reviewer’s suggestions, we have changed corresponding abbreviations, the parts of which were shown by blue highlights throughout the revised manuscript with remarks.

-----------------------------------------------------------------------

Comments on the Quality of English Language

The English is understandable, but it is mainly the form that needs improvement.

Avoid repetitions throughout the text (e.g. "at the age of 8 weeks", "without any visible skin inflammation").

Shorter sentences would greatly facilitate understanding.

Response:

According to the reviewer’s comments, we have modified corresponding sentences and deleted repetitions pointed out by the reviewer 2

-------------------------------------------------------------------------------------

Round 2

Reviewer 1 Report

Comments and Suggestions for Authors

This is one of the biggest problem of this work – the overexpression of enzyme that was shown to work as SM deacylase, does not show the activity. In addition to SPC, the authors should look at psychosine generation (glucosylsphingosine).

Yet, the fact that overexpression of seemingly inactive enzyme leads to AD-like symptoms in mice signifies that proper stratum corneum barrier is not formed. The authors suggest that this is due to an unknown function of SGDase. But the authors should also consider the following and discuss this in the manuscript.

Proper lamellae formation requires all the components, proteins and lipids, to be in the right proportion and properly assembled. SGDase is functioning in the released lamellae granules, outside of the cells, with the goal to create ceramides and allow their self-assembly into the lamellae. However, in TG-mice, an overly abundant SGDase (Fig 6, ASAH1beta) can potentially prohibit a proper lamellae assembly by interfering with biophysically-driven process of ceramide self-organization into the lamellae. In this case, overexpression of any acidic enzyme stored in the lamellae granules, and not only of ASHA1beta, may result in the same functional abnormality. The authors should at least consider such a possibility.

Author Response

To the Reviewer 1

Thank you very much again for your invaluable comments and suggestions.

We ‘d like to answer to your comments and suggestions as follows;

Reviewer 1

This is one of the biggest problem of this work – the overexpression of enzyme that was shown to work as SM deacylase, does not show the activity. In addition to SPC, the authors should look at psychosine generation (glucosylsphingosine).

Response:

We have already done the measurement of glucosylsphingosine as enzymatic reaction product by glucosylceramide deacylase and confirmed the absence of the activity although data were not described in the text of our manuscript.

---------------------------------------------------------------------------------------------------------------------

Yet, the fact that overexpression of seemingly inactive enzyme leads to AD-like symptoms in mice signifies that proper stratum corneum barrier is not formed. The authors suggest that this is due to an unknown function of SGDase. But the authors should also consider the following and discuss this in the manuscript.

Proper lamellae formation requires all the components, proteins and lipids, to be in the right proportion and properly assembled. SGDase is functioning in the released lamellae granules, outside of the cells, with the goal to create ceramides and allow their self-assembly into the lamellae. However, in TG-mice, an overly abundant SGDase (Fig 6, ASAH1beta) can potentially prohibit a proper lamellae assembly by interfering with biophysically-driven process of ceramide self-organization into the lamellae. In this case, overexpression of any acidic enzyme stored in the lamellae granules, and not only of ASHA1beta, may result in the same functional abnormality. The authors should at least consider such a possibility.

Response:

Thank you very much for such invaluable comments. Because we completely agree with this suggestion, we have added almost the same sentences in Discussion section as marked by blue highlight in the revised manuscript with remarks.

-------------------------------------------------------------------------------------

Reviewer 2 Report

Comments and Suggestions for Authors

Overall, the authors responded very briefly and made mainly formal changes to the manuscript. As a result, it remains very dense, with overly long sentences, making it difficult to read. The authors need to make a significant effort in writing to make their manuscript more understandable and less dense.

Contrary to what the authors claim, it cannot be stated that a defective epidermal barrier is not the cause of increased stress sensitivity in their TG mice. Apparently, the TG mice, under normal conditions, have a higher TEWL than the WT mice (Figure 2). It is likely that, as in non-lesional areas of AD patients, this is also accompanied by a disruption in the composition/organization of SC lipids. Therefore, it is important to verify this using a more rigorous technique than a simple immunohistological analysis (Figure 5b).

The authors suggest that the expression of the transgene is the cause of the "hypersensitivity" in the transgenic mice, regardless of ASAH1b enzymatic activity. However, they provide only very brief and undocumented explanations. The explanation about the organization of the lipid lamellae (second-to-last added paragraph in the discussion) is not clear. Would the mere overexpression of the enzyme, inactive, be enough to disorganize the lipids? Could this therefore be the case for any protein overexpressed in lamellar bodies? Furthermore, how can a high proportion of cysteines in an SC protein be a causal factor for increased sensitivity to stress? Has this been demonstrated? Published?

The explanation, which is very brief, justifying the absence of ASAH1b activity in TG mice (signal peptide?) is not understood.

Comments on the Quality of English Language

The authors need to make a significant effort in writing to make their manuscript more understandable and less dense.

Author Response

Responses to Reviewer 2

Thank you very much again for your invaluable comments and suggestions to further improve our manuscript. We responded to each of your comments and suggestions in a step-by-step manner as detailed below. Corrected, modified or added sentences are shown with blue highlighting or by Word tracking marks in the revised manuscript.

------------------------------------------------------------------------------------------------

Comment - Overall, the authors responded very briefly and made mainly formal changes to the manuscript. As a result, it remains very dense, with overly long sentences, making it difficult to read. The authors need to make a significant effort in writing to make their manuscript more understandable and less dense.

Response: According to your recommendation to make our manuscript more understandable and less dense, with help from an English-speaking scientific editing service, we modified many long sentences to make them more understandable and less dense throughout our revised manuscript. We attach a certificate of editing from that company for reference. Corrected, modified or added sentences are shown with blue highlighting or by Word tracking marks in the revised manuscript.

------------------------------------------------------------------------------------------------

Comment - Contrary to what the authors claim, it cannot be stated that a defective epidermal barrier is not the cause of increased stress sensitivity in their TG mice. Apparently, the TG mice, under normal conditions, have a higher TEWL than the WT mice (Figure 2). It is likely that, as in non-lesional areas of AD patients, this is also accompanied by a disruption in the composition/organization of SC lipids. Therefore, it is important to verify this using a more rigorous technique than a simple immunohistological analysis (Figure 5b).

Response: As mentioned earlier, because SGDase activity is not expressed in TG mice and since aCDase activity occurs at a similar level in TG and WT mice, we thought it likely that the mild barrier disruption could not be ascribed to the slight decrease in SC ceramide, but rather to an abnormal integrity of lipid lamellae due to the overly abundant ASAH1b. Therefore, ceramide quantitation by LC-MS/MS is not so important and an explanation of this was inserted in an appropriate position of the Discussion as marked with blue highlighting in the revised manuscript.

------------------------------------------------------------------------------------------------

Comment - The authors suggest that the expression of the transgene is the cause of the "hypersensitivity" in the transgenic mice, regardless of ASAH1b enzymatic activity. However, they provide only very brief and undocumented explanations. The explanation about the organization of the lipid lamellae (second-to-last added paragraph in the discussion) is not clear. Would the mere overexpression of the enzyme, inactive, be enough to disorganize the lipids? Could this therefore be the case for any protein overexpressed in lamellar bodies? Furthermore, how can a high proportion of cysteines in an SC protein be a causal factor for increased sensitivity to stress? Has this been demonstrated? Published?

Response: In light of the Reviewer’s comments, we deleted the text about the cysteine-rich ASAH1b and chemokine-like action, etc. Further, as mentioned above, the explanation about the organization of the lipid lamellae was modified to make it more understandable as shown with blue highlighting in the revised manuscript.

------------------------------------------------------------------------------------------------

Comment - The explanation, which is very brief, justifying the absence of ASAH1b activity in TG mice (signal peptide?) is not understood.

Response: As noted above, we have deleted the corresponding text about the signal peptides as shown with blue highlighting in the revised manuscript.

------------------------------------------------------------------------------------------------

Comments on the Quality of English Language

Comment - The authors need to make a significant effort in writing to make their manuscript more understandable and less dense.

Response: According to the Reviewer’s suggestion to make our manuscript more understandable and less dense, with help from an English-speaking scientist, we have modified many long sentences to make them more understandable and less dense throughout our revised manuscript. Corrected, modified or added sentences are shown with blue highlighting or by the Word tracking system in the revised manuscript.

------------------------------------------------------------------------------------------------

Round 3

Reviewer 2 Report

Comments and Suggestions for Authors

Overall, the authors responded very briefly and made mainly formal changes to the manuscript. As a result, it remains very dense, with overly long sentences, making it difficult to read. The authors need to make a significant effort in writing to make their manuscript more understandable and less dense.

Contrary to what the authors claim, it cannot be stated that a defective epidermal barrier is not the cause of increased stress sensitivity in their TG mice. Apparently, the TG mice, under normal conditions, have a higher TEWL than the WT mice (Figure 2). It is likely that, as in non-lesional areas of AD patients, this is also accompanied by a disruption in the composition/organization of SC lipids. Therefore, it is important to verify this using a more rigorous technique than a simple immunohistological analysis (Figure 5b).

The authors suggest that the expression of the transgene is the cause of the "hypersensitivity" in the transgenic mice, regardless of ASAH1b enzymatic activity. However, they provide only very brief and undocumented explanations. The explanation about the organization of the lipid lamellae (second-to-last added paragraph in the discussion) is not clear. Would the mere overexpression of the enzyme, inactive, be enough to disorganize the lipids? Could this therefore be the case for any protein overexpressed in lamellar bodies? Furthermore, how can a high proportion of cysteines in a SC protein be a causal factor for increased sensitivity to stress? Has this been demonstrated? Published?

The explanation, which is very brief, justifying the absence of ASAH1b activity in TG mice (signal peptide?) is not understood.

Comments on the Quality of English Language

The English is understandable, but it is mainly the form that needs improvement.

Shorter sentences would greatly facilitate understanding.

Author Response

Responses to Reviewer 2 (Round 3)

Response: Thank you very much again for your invaluable comments and suggestions to further improve our manuscript. We responded to each of your comments and suggestions in a step-by-step manner as detailed below. Corrected, modified or added sentences are shown with Word tracking marks in the revised manuscript.

------------------------------------------------------------------------------------------------

Comment -Overall, the authors responded very briefly and made mainly formal changes to the manuscript. As a result, it remains very dense, with overly long sentences, making it difficult to read. The authors need to make a significant effort in writing to make their manuscript more understandable and less dense.

Response: According to your recommendation to make our manuscript more understandable and less dense, with help from an English-speaking scientific editing service, we modified many long sentences to make them more understandable and less dense throughout our revised manuscript. We attach a certificate of editing from that company for reference. Corrected, modified or added sentences are shown by Word tracking marks in the revised manuscript.

-------------------------------------------------------------------------------------

Comment - Contrary to what the authors claim, it cannot be stated that a defective epidermal barrier is not the cause of increased stress sensitivity in their TG mice. Apparently, the TG mice, under normal conditions, have a higher TEWL than the WT mice (Figure 2). It is likely that, as in non-lesional areas of AD patients, this is also accompanied by a disruption in the composition/organization of SC lipids. Therefore, it is important to verify this using a more rigorous technique than a simple immunohistological analysis (Figure 5b).

Response: Almost all data were obtained from the dorsal skin of mice after hair removal using a depilatory cream containing glycolic acid. On the other hand, when the hair removal was performed using electric hair clippers, the dorsal skin of TG and WT mice at the age of 8 weeks did not have any visible skin symptoms including scaling, and they had similar TEWL values, which strongly suggests that there is no barrier disruption in the skin of TG mice under normal conditions, compared to WT mice. Therefore, we consider it appropriate to state that a defective epidermal barrier is not the cause of the increased stress sensitivity in TG mice. To make this point clearer, we have added new data in Figure 2 showing the appearance of the skin and TEWL values in TG and WT mice at the age of 8 weeks after hair removal using electric hair clippers. These additions are shown by the Word tracking system in the revised manuscript. In addition, because SGDase activity is not expressed in TG mice and since aCDase activity occurs at a similar level in TG and WT mice, we thought it likely that the mild barrier disruption could not be ascribed to the slight decrease in SC ceramide, but rather to an abnormal integrity of lipid lamellae due to the overly abundant ASAH1b. Therefore, ceramide quantitation by LC-MS/MS is not so important and an explanation of this was inserted in an appropriate position of the Discussion as marked by the Word tracking system in the revised manuscript.

-------------------------------------------------------------------------------------------------------------

Comment - The authors suggest that the expression of the transgene is the cause of the "hypersensitivity" in the transgenic mice, regardless of ASAH1b enzymatic activity. However, they provide only very brief and undocumented explanations. The explanation about the organization of the lipid lamellae (second-to-last added paragraph in the discussion) is not clear. Would the mere overexpression of the enzyme, inactive, be enough to disorganize the lipids? Could this therefore be the case for any protein overexpressed in lamellar bodies? Furthermore, how can a high proportion of cysteines in a SC protein be a causal factor for increased sensitivity to stress? Has this been demonstrated? Published?

Response: In light of the Reviewer’s comments, we deleted the text about the cysteine-rich ASAH1b and chemokine-like action, etc. Further, as mentioned above, the explanation about the organization of the lipid lamellae was modified to make it more understandable as shown by Word tracking system in the revised manuscript.

--------------------------------------------------------------------------------

Comment - The explanation, which is very brief, justifying the absence of ASAH1b activity in TG mice (signal peptide?) is not understood.

Response: We deleted the corresponding text about the signal peptides as shown by the Word tracking system in the revised manuscript.

--------------------------------------------------------------------------------------------------------------------
